# Perceived Preparedness of Dental Academic Institutions to Cope with the COVID-19 Pandemic: A Multi-Country Survey

**DOI:** 10.3390/ijerph18041445

**Published:** 2021-02-04

**Authors:** Nour Ammar, Nourhan M. Aly, Morenike Oluwatoyin Folayan, Yousef Khader, Simin Z. Mohebbi, Sameh Attia, Hans-Peter Howaldt, Sebastian Boettger, Jorma Virtanen, Marwa Madi, Diah A. Maharani, Anton Rahardjo, Imran Khan, Ola B. Al-Batayneh, Maher Rashwan, Verica Pavlic, Smiljka Cicmil, Kanako Noritake, Gabriella Galluccio, Antonella Polimeni, Anas A. Shamala, Arheiam Arheiam, Davide Mancino, Prathip Phantumvanit, Jin-Bom Kim, Youn-Hee Choi, Mai A. Dama, Maha M. Abdelsalam, Jorge L. Castillo, Myat Nyan, Iyad Hussein, Easter Joury, Ana P. Vukovic, Alfredo Iandolo, Arthur M. Kemoli, Maha El Tantawi

**Affiliations:** 1Department of Pediatric Dentistry and Dental Public Health, Faculty of Dentistry, Alexandria University, Alexandria 21253, Egypt; Nourhan.moustafa@alexu.edu.eg (N.M.A.); maha.tantawy@alexu.edu.eg (M.E.T.); 2Department of Child Dental Health, Obafemi Awolowo University, Ile-Ife 22005, Nigeria; toyinukpong@yahoo.co.uk; 3Department of Public Health, Jordan University of Science and Technology, Irbid 22110, Jordan; yskhader@just.edu.jo; 4Research Center for Caries Prevention, Dentistry Research Institute, Tehran University of Medical Sciences, Tehran 14399-55991, Iran; smohebbi@tums.ac.ir; 5Community Oral Health Department, School of Dentistry, Tehran University of Medical Sciences, Tehran 14399-55991, Iran; 6Department of Cranio-Maxillofacial Surgery, Justus-Liebig University Giessen, 35392 Giessen, Germany; Sameh.Attia@dentist.med.uni-giessen.de (S.A.); hp.howaldt@uniklinikum-giessen.de (H.-P.H.); sebastian.boettger@uniklinikum-giessen.de (S.B.); 7Department of Clinical Dentistry, Faculty of Medicine, University of Bergen, 5020 Bergen, Norway; jorma.virtanen@uib.no; 8Department of Preventive Dental Sciences, College of Dentistry, Imam Abdulrahman Bin Faisal University, Dammam 34221, Saudi Arabia; mimadi@iau.edu.sa; 9Department of Preventive and Public Health Dentistry, Faculty of Dentistry, Universitas Indonesia, 16424 Depok, Indonesia; diah.ayu64@ui.ac.id (D.A.M.); antonrahardjo@gmail.com (A.R.); 10Department of Oral & Maxillofacial Surgery, Faculty of Dentistry, Jamia Millia Islamia, New Delhi 110025, India; ikhan3@jmi.ac.in; 11Department of Preventive Dentistry, Faculty of Dentistry, Jordan University of Science and Technology, Irbid 22110, Jordan; olabt@just.edu.jo; 12Center for Oral Bioengineering, Barts and the London, School of Medicine and Dentistry, Queen Mary University of London, Mile End Road, London E1 4NS, UK; m.r.a.mohamed@qmul.ac.uk; 13Department of Conservative Dentistry, Faculty of Dentistry, Alexandria University, Alexandria 21253, Egypt; 14Department of Periodontology and Oral Medicine, Institute of Dentistry, 78000 Banja Luka, Bosnia and Herzegovina; verica.pavlic@med.unibl.org; 15Department of Oral Rehabilitation, Faculty of Medicine Foca, University of East Sarajevo, 73300 Foca, Bosnia and Herzegovina; smiljka.cicmil@ues.rs.ba; 16Oral Diagnosis and General Dentistry Department, Dental Hospital, Tokyo Medical and Dental University, Tokyo 113-8510, Japan; noritake.irm@tmd.ac.jp; 17Department of Oral and Maxillo Facial Sciences, Faculty of Medicine and Dentistry, Sapienza University of Rome, 00185 Rome, Italy; gabriella.galluccio@uniroma1.it (G.G.); antonella.polimeni@uniroma1.it (A.P.); 18Department of Preventive and Biomedical Science, Faculty of Dentistry, University of Science and Technology, Sanaa 15201, Yemen; anasshamala@gmail.com; 19Department of Community and Preventive Dentistry, Faculty of Dentistry, University of Benghazi, Benghazi, Libya; arheiam@gmail.com; 20Department of Endodontics and Conservative Dentistry, Faculty of Dental Medicine, University of Strasbourg, 67000 Strasbourg, France; Davidemancino@icloud.com; 21Department of Biomaterials and Bioengineering, INSERM UMR_S 1121, Strasbourg University, 67000 Strasbourg, France; 22Faculty of Dentistry, Thammasat University, Bangkok 10200, Thailand; prathipphan@gmail.com; 23Department of Preventive and Community Dentistry, School of Dentistry, Pusan National University, Yangsan 50612, Korea; jbomkim@pusan.ac.kr; 24Department of Preventive Dentistry, School of Dentistry, Kyungpook National University, Daegu 41940, Korea; cyh1001@knu.ac.kr; 25Orthodontics and Pediatric Dentistry Department, Faculty of Dentistry, Arab American University, Jenin 240, Zababdeh 13, Palestine; mai.dama@aaup.edu; 26Department of Biomedical Dental Sciences, College of Dentistry, Imam Abdulrahman Bin Faisal University, Dammam 34221, Saudi Arabia; mmabdelsalam@iau.edu.sa; 27Department of Dentistry for Children and Adolescents, Universidad Peruana Cayetano Heredia, Lima 15102, Peru; jorge.castillo@upch.pe; 28Department of Prosthodontics, University of Dental Medicine, Mandalay 05041, Myanmar; myatnyan@gmail.com; 29Department of Pediatric Dentistry, Mohammed Bin Rashid University of Medicine and Health Sciences, Dubai 800 MBRU (6278), United Arab Emirates; Iyad.Hussein@mbru.ac.ae; 30Centre for Dental Public Health and Primary Care, Institute of Dentistry, Barts and The London School of Medicine and Dentistry, Queen Mary University of London, London E1 2AD, UK; e.joury@qmul.ac.uk; 31Department of Pediatric and Preventive Dentistry, School of Dental Medicine, University of Belgrade, Beograd, 11000 Belgrade, Serbia; ana.vukovic@stomf.bg.ac.rs; 32Department of Endodontics, University of Salerno, 84080 Fisciano, Italy; iandoloalfredo@libero.it; 33Department of Paediatric Dentistry & Orthodontics, School of Dental Sciences, University of Nairobi, Nairobi 30197-00100, Kenya; musakulu@gmail.com

**Keywords:** COVID-19, academics, dental, surveys and questionnaires, pandemic, multilevel analysis, institution, preparedness

## Abstract

Dental academic institutions are affected by COVID-19. We assessed the perceived COVID-19 preparedness of these institutions and the characteristics of institutions with greater perceived preparedness. An international cross-sectional survey of dental academics was conducted from March to August 2020 to assess academics’ and institutional attributes, perceived preparedness, and availability of infection prevention and control (IPC) equipment. Principal component analysis (PCA) identified perceived preparedness components. Multilevel linear regression analysis assessed the association between perceived preparedness and fixed effect factors (academics’ and institutions’ attributes) with countries as random effect variable. Of the 1820 dental academics from 28 countries, 78.4% worked in public institutions and 75.2% reported temporary closure. PCA showed five components: clinic apparel, measures before and after patient care, institutional policies, and availability of IPC equipment. Significantly less perceived preparedness was reported in lower-middle income (LMICs) (B = −1.31, *p* = 0.006) and upper-middle income (UMICs) (B = −0.98, *p* = 0.02) countries than in high-income countries (HICs), in teaching only (B = −0.55, *p* < 0.0001) and in research only (B = −1.22, *p* = 0.003) than teaching and research institutions and in institutions receiving ≤100 patients daily than those receiving >100 patients (B = −0.38, *p* < 0.0001). More perceived preparedness was reported by academics with administrative roles (B = 0.59, *p* < 0.0001). Academics from low-income countries (LICs) and LMICs reported less availability of clinic apparel, IPC equipment, measures before patient care, and institutional policies but more measures during patient care. There was greater perceived preparedness in HICs and institutions with greater involvement in teaching, research, and patient care.

## 1. Introduction

In December 2019, the identification of a highly infectious respiratory disease in China raised international public health concerns. The COVID-19 disease was caused by the SARS-CoV-2 coronavirus and was declared a pandemic by the World Health Organization (WHO) on 11 March 2020 [1]. As of 1 October 2020, there were 33.8 million confirmed COVID-19 cases and one million deaths [2]. The pandemic has had an impact on various sectors of life and the economy. In several countries, the operations of universities and schools—among many other institutions—have been severely affected.

Dental schools were faced with multiple challenges during the pandemic [3]. They had to balance the implementation of a stay-at-home policy and the suspension of clinical teaching with the provision of emergency services to patients [4,5] at the same time, and continuing quality dental education [6] through the use of technology for distance teaching and learning [7,8,9]. For dental schools in developing countries, the cost of shifting to online education was huge, preventing many schools from making the needed transition. The shutdown of dental academic institutions because of COVID-19 also raised students’ concerns regarding the replacement of hands-on-experience with didactic online teaching, uncertainty about graduating during the pandemic, and clinical competence at graduation [7].

The COVID-19 pandemic, therefore, challenged and continues to challenge dental academic institutions by testing the limits of their emergency preparedness, elevating the stress levels of dental academics [10], and requiring them to be up to date with the latest knowledge of the characteristics and transmissibility of this disease [11]. Some academic institutions were more prepared than others to establish emergency response plans that made it possible to return to near normalcy for dental students’ education and reduced the health and safety risks for faculty, staff, and students. 

Preparedness refers to the ability to anticipate and respond effectively to the impact of hazards [12]. Assessing various aspects of dental academic institutions’ preparedness to deal with the pandemic is important to prepare for future operational scenarios and balance the pros and cons of continued closure versus opening and resuming work. Understanding the characteristics associated with different degrees of preparedness helps prioritize institutions with a greater need for support and makes it possible for academics and other stakeholders to address weaknesses and have realistic expectations for the short-term and intermediate future of dental education in addition to investing in human and non-human resources needed for supporting measures such as online education and clinical placement. 

The aim of the study was to assess the various aspects of institutional preparedness perceived by dental academics in response to the COVID-19 pandemic in several countries around the world and to determine the institutional characteristics associated with perceived preparedness. The null hypothesis of the study was that perceived preparedness would not be associated with institutional characteristics and would not differ by country. 

## 2. Methods

A cross-sectional survey of dental academics was conducted using an online questionnaire from March to August 2020. Ethical approval for the study was obtained from the Research Ethics Committee of the Faculty of Dentistry, Alexandria University, Alexandria, Egypt (IRB 00010556-IORG 0008839 /6-11-2016). 

We tried to identify a sampling frame for dental academic institutions to calculate sample size and plan sampling strategy. However, except for North America [13] and the European Union [14], a comprehensive list of these institutions in other countries/regions could not be identified. In addition, it was not possible to identify emails of deans of targeted institutions so that they could represent their institutions. We used dental academics’ views and perceptions to assess institutional preparedness and the unit of sampling and analysis was the academic. Academics were invited if they worked in a dental education or research institution, had access to the Internet, could read and understand the languages of the survey, and if they consented to participate. No restrictions were imposed by specialty or academic degree. Non-academic staff and students were excluded. The total sample size was calculated using an online calculator [15] at 80% power and 95% confidence level. To maximize the calculated sample size, it was assumed that 50% of academics would report preparedness in their institutions. The sample size needed to estimate this proportion within a margin error of 0.05 was 383. In addition, to address the study aim related to the association between characteristics and preparedness, we planned for power to detect a small effect size of 0.2 with 5% alpha error and 80% study power and calculated that 788 participants would be needed [16]. We continued data collection after this number was reached to maximize geographic representativeness in addition to study power. 

### 2.1. Questionnaire 

The questionnaire (Appendix A.1) was developed based on guidelines from the WHO, the Centers for Disease Control and Prevention (CDC), and previous research [17,18,19]. We followed the framework of the hierarchy of controls [20] used to address occupational hazards and protect workers. This hierarchy includes elimination and substitution which were not feasible regarding COVID-19 at the time the study was planned. The hierarchy also includes engineering controls, administrative controls, and personal protective equipment. We used these resources after adaptation to the situation of dental academic institutions during the pandemic, to develop the questionnaire which included questions that assessed various aspects of perceived preparedness; including the presence of supplies and equipment to control the pandemic and the spread of infection [17], how care is provided to patients to reduce the risk of infection [19], and institutionalizing these changes so that they are part of the policies and procedures of the academic institution [9]. The questionnaire was assessed for content validity by five dental academics who were not involved in the study. Their responses were not included in the final data analysis. The content validity index was calculated to be 0.99 [21]. 

The questionnaire consisted of three sections. The first section included one question where participants reported on the presence of 15 preparedness measures in their institutions based on their own perception. These measures would be applied if the institution was completely open or if only its emergency facilities were open during the pandemic. They described special precautions before and during patient care in addition to changes in policies to adapt to the pandemic situation. The items were a list where multiple selections were allowed. 

The second section included a question assessing the perceived availability of infection prevention and control (IPC) equipment. This was a list of 14 items with multiple selections allowed. 

The third section assessed the characteristics of dental academic institutions reported by academics such as: type of institution (public or private), whether it was an institution for teaching, research, or both, type of program (undergraduate, postgraduate, or both), size of the institution in terms of the total number of students and staff (categorized to up to 400 persons and more than 400 persons), number of patients receiving care per day (categorized into up to 100 patients and more than 100 patients), and whether the institution was temporarily closed because of the pandemic. We did not collect information about the identity of the institution to maintain confidentiality for ethical considerations. This section also assessed whether participants had an administrative role in the institution in which case they may have been more aware of existing preparedness measures. 

The questionnaire was preceded by a brief introduction explaining the purpose of the study, assuring voluntary participation, that participants could leave the survey at any point without incurring any penalties, and that their responses were confidential, anonymous, and accessible only to the study team. The questionnaire was uploaded to the electronic survey platform—SurveyMonkey—and took an average of five minutes to complete. It was originally developed in English and translated into Farsi by a co-author from Iran (SM), and into Portuguese by a dentist in the private sector from Brazil who was not further involved in the study (TSC), then translated again to English using back-translation to compare differences between the English and translated versions and ensure accuracy. All versions had the same questions with the same options for responses in the same sequence. The translations ensured that the translated versions used cultural and scientific terms that were proper for the local context. 

A call for collaborators was posted on ResearchGate and sent by email to invite researchers from several countries to manage participants’ recruitment in their own countries. Country collaborators received the study proposal explaining the sampling strategy, methods, timeline, and other details, in addition to customized survey links to use in their respective countries. They also received the approval of the Ethics Committee from Alexandria University to support their activities or use it to apply for ethical approvals in their own institutions. Convenience and snowball sampling strategies were used to distribute these links to the collaborators’ eligible contacts, post them on social media groups specific to dental academics and send them to institutional and network email lists. Distribution with maximum geographic coverage per country was attempted to the greatest extent possible and as much as the branching and decentralized nature of social media would allow. Two reminders were sent to the participants after 5 and 7 weeks.

### 2.2. Analysis

Data were described using frequencies and percentages. Kaiser–Meyer–Olkin measure for all 29 items—reflecting the measures of perceived preparedness in the 1st and 2nd sections of the survey—was calculated to be 0.94 and the *p*-value of Bartlett’s test of sphericity was statistically significant (*p* < 0.0001) supporting the use of principal component analysis (PCA) to identify the components within the preparedness measures [22]. The extraction of components was based on eigenvalues >1. Varimax rotation with Kaiser normalization was used and loading coefficients <0.35 were suppressed to show factor loadings with adequate strength and allow the interpretation of components and their factors. Regression coefficients of the factors were saved to the dataset and used for further analysis. Based on these coefficients, components and total preparedness scores were calculated as the sum of the regression coefficients of the factors. The mean of these scores was zero with positive regression coefficients (with values above zero) increasing the preparedness score and indicating greater preparedness [23].

A multilevel linear regression model was used to assess the characteristics associated with total perceived preparedness score as the dependent variable and account for the clustering of participants within countries. The independent variables were fixed effect factors reported by participants in the questionnaire: characteristics of the academic institution, and whether the participant had an administrative role. In addition, country income level was also used as a fixed effect factor. Income was classified according to the World Bank Databank [24] based on per capita gross national income (GNI) into low-income countries (LICs) with GNI <1026 USD, lower middle-income countries (LMICs) with GNI between USD 1026 and 3995, upper middle income countries (UMICs) with GNI between USD 3996 and 12,375, and high income countries (HICs) with GNI >12,375 USD. Country of residence was used as a random effect factor since it could confound the association between perceived preparedness and the explanatory variables. Multivariate analysis of variance (MANOVA) was used to assess the association between country-level income and the components (measures of preparedness) identified in the PCA which served as dependent variables in this model adjusting for institutional characteristics. Regression coefficients, 95% confidence intervals (CIs), and *p* values were calculated. 

## 3. Results

There were 1820 complete questionnaires from participants in 28 countries (Appendix A.2: Countries included in the study). Table 1 shows the characteristics of participants and their academic institutions and countries. Most participants were from affluent countries (715; 39.3% from UMICs and 691; 38.0% from HICs) and 571 (31.4%) were from Eastern Mediterranean countries. Most (78.4%) participants worked in public institutions and 51% had administrative roles. In addition, 75.2% reported that their institutions were temporarily closed because of the pandemic. There were significantly less (*p* < 0.0001) reports of institutional closure from HICs (63%) than LICs (87.2%), LMICs (74.7%), and UMICs (86.6%). 

The percentages of academics who reported on various aspects of perceived preparedness measures are shown in Table 2. The most frequent perceived preparedness measures related to personal apparel used in the clinics were eye protection equipment (76.0%) and the least frequent were boots and closed work shoes (28.7%). The most frequent measures adopted before care provision was denying elective procedures (66.6%) and the least frequent was enforcing the isolation of persons suspected of having COVID-19 in the waiting area (40%). Measures adopted during care provision were the least frequent among all preparedness measures and included dedicating dental units (31.7%), personnel (22.9%), or instruments (21.9%) to treat COVID-19 suspected or infected persons. The most common institutional policy measures were postponing or canceling events (85.4%), and the least frequent were making sick leave policies more flexible (36.6%). The most available IPC equipment was rubber dam isolation kits (64.7%) and the least available was 4-handed dentistry support (48.2%). 

Table 2 also shows the results of the PCA for perceived preparedness measures. Factor loadings revealed that there were five components describing perceived preparedness reported by the participants: availability of personal apparel for clinic use (5 factors), measures taken before providing clinical care to patients (7 factors), measures followed during care provision (4 factors), institutional policies (5 factors), and availability of IPC equipment (5 factors). These five components collectively explained 50.9% of the variation in responses among participants with a percentage of variation explained by individual components ranging from 8.3% for the availability of IPC equipment to 11.3% for the availability of personal apparel. Factor loadings ranged from 0.37 to 0.83 and were positive indicating that participants who reported these measures were likely to be describing greater preparedness.

Table 3 shows that in unadjusted analysis, the association was significant between overall perceived preparedness and income level (*p* = 0.045, 0.002 and 0.005), type of institution (*p* = 0.032), whether it was a teaching only (*p* < 0.0001) or a research only institution (*p* = 0.009), the number of patients served (*p* < 0.0001), and having an administrative role (*p* < 0.0001). In adjusted analysis, the findings show that participants reported significantly less overall perceived preparedness in LMICs (B = −1.31, 95% CI: −2.25, −0.37) and UMICs (B = −0.98, 95% CI: −1.78, −0.17) than HICs and in teaching only (B = −0.55, 95% CI: −0.81, −0.28) and research only (B = −1.22, 95% CI: −2.01, −0.43) than teaching and research institutions. Significantly less perceived preparedness was reported in institutions receiving ≤100 patients daily than those receiving >100 patients (B = −0.38, 95% CI: −0.59, −0.18). Participants with an administrative role reported significantly more perceived preparedness (B = 0.59, 95% CI: 0.40, 0.78). 

Participants from LICs, LMICs, and UMICs reported significantly less perceived availability of clinic apparel (B = −0.62, −0.81 and −0.31, *p* < 0.0001), less precautionary measures before patient care (B = −0.66, −0.23 and −0.44), *p* < 0.0001), and less institutional policies than participants from HICs (B = −0.65, −0.33 and −0.59, *p* < 0.0001, Table 4). In addition, participants from LMICs and UMICs reported significantly less availability of IPC equipment than participants from HICs (B = −0.23 and −0.33, *p* < 0.0001). On the other hand, participants from LICs reported significantly more precautionary measures during patient care than those from HICs (B = 0.59, *p* < 0.0001).

## 4. Discussion

To our knowledge, this is the first study that identified dental academic institutional characteristics associated with dental academics’ perceived preparedness for COVID-19. This finding is important to inform the prioritization of responses to support academic institutions with emergency plans during epidemics like COVID-19. Income levels of countries were significantly associated with the degree of perceived preparedness among participants. Perceived preparedness was also associated with the activities of the academic institution such as teaching, research, and patient care. Participants who had administrative roles reported more perceived pandemic preparedness. The null hypothesis of the study is, thus, rejected.

The study findings have multiple implications for emergency preparedness policies at country and institutional levels. First, education systems and monitoring criteria are expected to vary among countries. However, it is important that regulatory bodies ensure the consistent application of essential safety measures and preparedness policies in various academic institutions: those involved in teaching or research and those serving a large number of patients or smaller institutions. This may also be supported by establishing exchange programs between dental academic institutions to train staff and share experiences to improve preparedness for emergencies. Collaborative efforts to manage the impact of the COVID-19 pandemic are important within and between institutions.

Second, the perceived availability of IPC equipment and personal apparel may affect dental education, just as it affects healthcare services provision [25]. Dentists providing dental care are among the healthcare workers who are most exposed to infection [19]. In addition, dental academic institutions are visited and used by a substantial number of people including academics, students, patients, assistants, and administrative staff. Inherently, it is a major challenge to protect the health of all these individuals and at the same time, provide proper dental care, and ensure high-quality education [7]. Concern about the risk of contracting infection due to the unavailability of IPC equipment and personal apparel may be a valid reason for temporarily shutting down dental schools if they are less prepared for patient care.

There are several recommendations on how to deal with COVID-19 including guidelines from the CDC [17], the WHO [2], the National Health Service UK (NHS) [26], professional and academic dental associations [27,28,29], and experts [7,30]. While following these recommendations and applying special precautions before, during, and after patient care may minimize the risk of infection [19,31,32], the competency to apply these guidelines may be difficult to achieve. Dental educators may benefit from reflecting on how other institutions were able to reduce the risk of infection during this pandemic and use the lessons learned to revise institutional policies on emergency preparedness plans. 

Academics who were administrators reported more perceived preparedness measures. This may be biased reporting, reflecting their higher concern to ensure patient safety and minimize the possibility of cross-infection [32] in dental clinics during the pandemic in addition to their own perception that they are expected to report more preparedness. Alternatively, it may be attributed to their greater awareness of the institutional infrastructure and processes. This emphasizes the need to seek the views of different stakeholders within the same institution and not only those in administrative positions when assessing institutional preparedness. Such bias may have significant implications for establishing appropriate support programs.

Perceived preparedness for COVID-19 was worse in less affluent countries than in HICs, resulting in more schools being temporarily closed in less affluent countries. This may lead to students’ inability to have hands-on-experiences in providing dental care during emergencies, in addition to problems related to inadequate clinical training in general and with concern about the quality of dental education in less affluent countries during the COVID-19 period. This finding demonstrates that international support for dental education should prioritize institutions in less affluent countries by providing human, financial, and technical resources to ensure quality online schooling, share teaching resources and best practices, and train academics to build capacity to respond to emergencies. The challenge that less affluent countries face becomes greater because of the economic recession caused by COVID-19 [33,34] and its impact on job opportunities in academia [35] and on financial support for students with an effect on the number of enrolled students [36]. Dental academic institutions in less affluent countries had to struggle to provide acceptable levels of dental education before the pandemic [35] and the pandemic may have made these constraints a lot harder to overcome.

One of the strengths of the current study is the substantial number of dental academics from many countries representing various income levels and geographic regions. The study also captured information on perceived preparedness plans in different education systems. However, the study had some limitations. Due to its cross-sectional design, perceived preparedness was recorded at one point in time and further changes in some institutions may have occurred after data collection. Furthermore, during the period of data collection, different countries may have been exposed to different stages of the pandemic with different responses. Because of these variations, we included the country of residence as a random effect variable in the multilevel model to account for differences among countries. In addition, the study design does not support causality and can only suggest associations and therefore, the direction of associations cannot be ascertained. For example, institutions receiving up to 100 patients had less perceived preparedness and this may be attributed to regulatory measures reducing the number of patients in less prepared institutions or, alternatively, to institutions receiving fewer patients having less incentive to ensure preparedness. Some under-reporting may have occurred because non-administrators were unaware of the ongoing preparations to protect personnel and patients especially when the institutions were closed. It was not possible to target the deans of dental schools because no information about them could be retrieved outside North America and Europe. We sent the survey, instead, to dental academics whose contact information was available and, therefore, it is possible that more than one participant per institution might have responded. Whether their responses agreed or differed could not be ascertained because we did not collect information about the institution’s identity for ethical considerations. We used multilevel analysis to accommodate the clustering of participants in the countries that we identified. This might have addressed—although not fully—their clustering in institutions which are clustered within countries. We also controlled for institutional characteristics that may be associated with perceived preparedness. Because of the limitation imposed by the sampling strategy and the ethical concerns, the study was, thus, limited to assessing perceived preparedness. Despite these limitations, the study provides a snapshot of the preparedness measures adopted in dental academic institutions in several countries and this may help in designing adequate plans to address COVID-19 and similar outbreaks in the future. Future studies are needed [37] to assess developments in perceived and actual preparedness based on reports of academics and administrators in various institutions as the pandemic develops.

## 5. Conclusions

Perceived preparedness to cope with COVID-19 varied among dental academics with better perceived preparedness reported by academics who had administrative positions, those working in teaching and research institutions rather than those working in institutions involved in either one exclusively, and those working in institutions receiving a large number of patients. Academics from HICs and less affluent countries indicated different preparedness strategies to cope with the pandemic.

## Figures and Tables

**Table 1 ijerph-18-01445-t001:** Characteristics of countries, academic institutions, and participants included in the study (n = 1820).

Factor		N (%)
Country characteristics		
Income level	LICs	39 (2.1)
	LMICs	375 (20.6)
	UMICs	715 (39.3)
	HICs	691 (38.0)
Region	Africa	54 (3.0)
	Americas	309 (17.0)
	South-East Asia	408 (22.4)
	Europe	400 (22.0)
	Eastern Mediterranean	571 (31.4)
	Western Pacific	78 (4.3)
Dental academic institution characteristics		
Type	Public	1426 (78.4)
	Private	394 (21.6)
Size	1–400 persons	781 (42.9)
	>400 persons	1039 (57.1)
Mission	Teaching only	296 (16.3)
	Research only	24 (1.3)
	Teaching and research	1500 (82.4)
Programs offered	Undergraduate	290 (15.9)
	Postgraduate	81 (4.5)
	Under and postgraduate	1449 (79.6)
Number of patients received daily	0–100 patients	767 (42.1)
	>100 patients	1053 (57.9)
Institution temporarily closed because of the pandemic	Yes	1368 (75.2)
	No	452 (24.8)
Participant characteristics		
Has administrative role	Yes	928 (51)
	No	892 (49)

LICs: low-income countries, LMICs: lower-middle income countries, UMICs: upper-middle income countries, and HICs: high-income countries.

**Table 2 ijerph-18-01445-t002:** Principal component analysis (PCA) for perceived preparedness of dental academic institutions.

Items	N (%)	Factor Loadings				
		Personal Apparel	Before Patient Care	During Patient Care	Institutional Policies	Infection Control Equipment
Percentage of variance explained	-	11.3%	10.9%	10.3%	10.1%	8.3%
Install triage to screen for symptoms	909 (49.9)		0.59			
Deny elective procedures to patients with COVID-19	1212 (66.6)		0.69			
Ensure a distance of at least 1 meter in the waiting area	927 (50.9)		0.55			
Post visual signs for hygiene	1078 (59.2)		0.50			
Ask questions about respiratory symptoms in medical history	903 (49.6)		0.55			
Ask patients to call about respiratory symptoms before visit	759 (41.7)		0.44			
Provide training on infection control guidelines	598 (32.9)		0.37			
Enforce isolation of COVID-19 patients in waiting area	728 (40)			0.49		
Dedicate personnel to treat only COVID-19 patients	416 (22.9)			0.79		
Dedicate instruments to treat only COVID-19 patients	398 (21.9)			0.83		
Dedicate dental units to treat only COVID-19 patients	577 (31.7)			0.71		
Change sick leave policies to be flexible for affected staff	667 (36.6)				0.55	
Encourage home isolation of those who traveled abroad	1141 (62.7)				0.60	
Constitute COVID-19 preparedness and response committee	761 (41.8)				0.59	
Temporarily postpone or cancel events	1555 (85.4)				0.61	
Develop an emergency communication plan	899 (49.4)				0.65	
Availability of high-volume saliva ejectors	1082 (59.5)					0.75
Availability of rubber dam isolation	1177 (64.7)					0.70
Availability of preoperative anti-microbial mouth rinse	1073 (59.0)					0.49
Availability of extra-oral instead of intra-oral radiographs	993 (54.6)					0.59
Availability of 4-handed dentistry support	877 (48.2)					0.46
Availability of respirator N95 or FFP2 standard or equivalent	809 (44.5)	0.64				
Availability of long-sleeved water-resistant gown	975 (53.6)	0.58				
Availability of eye protection equipment	1384 (76.0)	0.61				
Availability of head cap	1165 (64.0)	0.65				
Availability of boots or closed work shoes	523 (28.7)	0.69				

**Table 3 ijerph-18-01445-t003:** Multilevel linear regression analysis for factors affecting overall perceived preparedness in dental academic institutions.

Variables		Unadjusted Estimates		Adjusted Estimates	
		B (95% CI)	*p* Value	B (95% CI)	*p* Value
Country income level	LICs	−1.62 (−3.21, −0.04) *	0.045 *	−1.42 (−3.00, 0.17)	0.08
	LMICs	−1.49 (−2.43, −0.56) *	0.002 *	−1.31 (−2.25, −0.37)	0.006 *
	UMICs	−1.15 (−1.96, −0.35) *	0.005 *	−0.98 (−1.78, −0.17)	0.02 *
	HICs	Reference category		Reference category	
Type of institution:	Public	0.29 (0.03, 0.55) *	0.032 *	0.21 (−0.05, 0.46)	0.119
	Private	Reference category		Reference category	
Size of institution	1–400 persons	−0.11 (−0.300, 0.088)	0.283	0.04 (−0.16, 0.24)	0.698
	>400 persons	Reference category		Reference category	
Focus of institution	Teaching only	−0.68 (−0.94, −0.43) *	<0.0001 *	−0.55 (−0.81, −0.28)	<0.0001 *
	Research only	−1.06 (−1.85, −0.26) *	0.009*	−1.22 (−2.01, −0.43)	0.003 *
	Teaching and research	Reference category		Reference category	
Type of program	Undergraduate only	−0.21 (−0.47, 0.06)	0.122	0.06 (−0.21, 0.33)	0.678
	Postgraduate only	0.14 (−0.32, 0.59)	0.554	0.37 (−0.08, 0.83)	0.108
	Undergraduate and postgraduate	Reference category		Reference category	
Number of patients served daily	0–100 patients	−0.46 (−0.66, −0.27) *	<0.0001 *	−0.38 (−0.59, −0.18)	<0.0001 *
	>100 patients	Reference category		Reference category	
Having administrative role	Yes	0.63 (0.45, 0.82) *	<0.0001 *	0.59 (0.40, 0.78) *	<0.0001 *
	No	Reference category		Reference category	

LICs: low-income countries, LMICs: lower-middle income countries, UMICs: upper-middle income countries, and HICs: high-income countries. B: regression coefficient, CI: confidence interval, *: statistically significant when *p* < 0.05. The country of residence is included as a random effect factor.

**Table 4 ijerph-18-01445-t004:** Association between country level income and components of perceived preparedness reported by dental academics in dental academic institutions, using MANOVA.

Variables	LICs¶		LMICs¶		UMICs¶	
	B (95% CI)	*p* Value	B (95% CI)	*p* Value	B (95% CI)	*p* Value
Personal apparel	−0.62 (−0.94, −0.31)	<0.0001 *	−0.81 (−0.94, −0.68)	<0.0001 *	−0.31 (−0.42, −0.21)	<0.0001 *
Before patient care	−0.66 (−0.98, −0.34)	<0.0001 *	−0.23 (−0.36, −0.10)	<0.0001 *	−0.44 (−0.54, −0.33)	<0.0001 *
During patient care	0.59 (0.26, 0.91)	<0.0001 *	0.10 (−0.03, 0.23)	0.13	−0.01 (−0.12, 0.10)	0.83
Institutional policies	−0.65 (−0.97, −0.33)	<0.0001 *	−0.33 (−0.46, −0.20)	<0.0001 *	−0.59 (−0.69, −0.48)	<0.0001 *
Infection control equipment	−0.32 (−0.64, 0.001)	0.05	−0.23 (−0.36, −0.10)	<0.0001 *	−0.33 (−0.44, −0.23)	<0.0001 *

LICs: low-income countries, LMICs: lower-middle income countries, UMICs: upper-middle income countries, and HICs: high-income countries. ¶: the reference category is HICs. B: regression coefficient, CI: confidence interval, *: statistically significant when *p* < 0.05.

## Data Availability

The data presented in this study are available on request from the corresponding author. The data are not publicly available as they will be used for further analysis.

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
