# Peer review of "Perceived Preparedness of Dental Academic Institutions to Cope with the COVID-19 Pandemic: A Multi-Country Survey"

_ijerph, 2021, doi:10.3390/ijerph18041445_

Round 1

Reviewer 1 Report

The authors made significant changes that improved the manuscript. There are very few corrections that need to be made before the manuscript is published. In addition, I suggest re-formatting the tables so that they become clearer.

Corrections were made in the pdf file (attached).

Figure 1 appears as supplemental material, however it is not mentioned in the text. Is that correct?

Author Response

We thank the Editor and the Reviewers for the opportunity to revise our paper and the constructive comments to improve it. We provide below a point-by-point response to all comments.

Reviewer comments:

  • The authors made significant changes that improved the manuscript. There are very few corrections that need to be made before the manuscript is published. In addition, I suggest re-formatting the tables so that they become clearer.

Response: We reformatted the tables as grids based on the Reviewer’s suggestion so that they are clearer and easier to follow.

  • Figure 1 appears as supplemental material, however it is not mentioned in the text. Is that correct?

Response: We thank the Reviewer for bringing this issue to our attention. The content of Figure 1 is now represented in Table 2 and in the text lines 300-311. It belonged to a previous version of the paper which we did not remove from the system. The reference to Figure 1 has now been removed.  

  • Corrections were made in the pdf file (attached).

Response: We have applied the Reviewer’s corrections to the currently revised manuscript.

  • Which author/institution translated the questionnaire in Brazil?

Response: The questionnaire was translated by Dr. Thiago Sant’Anna Coutinho from the Instituto Nacional de Cardiologia, Rio de Janeiro, Brazil. After he translated the questionnaire, he was not further involved in the study and is, therefore, not an author. We added a brief of this detail in the Methods section of the manuscript and included him in the Acknowledgements.

Reviewer 2 Report

While the authors have explained their approach and the reasons why they chose it, plus various mitigating factors that indeed made the performance of this study more complex, the fundamental issue I previously raised has not been addressed. This remains highly problematic for being able to interpret and understand the results.

Author Response

We thank the Editor and the Reviewers for the opportunity to revise our paper and the constructive comments to improve it. We provide below a point-by-point response to all comments.

Reviewer comments:

  • While the authors have explained their approach and the reasons why they chose it, plus various mitigating factors that indeed made the performance of this study more complex, the fundamental issue I previously raised has not been addressed. This remains highly problematic for being able to interpret and understand the results.

Response: We assumed the Reviewer’s concern is about the way we have presented the data. Based on this assumption, we have revised the manuscript to show with more clarity that the analysis represents the academics’ perception of the preparedness of their dental schools for the pandemic. In our case, the academics and not the dental schools are the units of analysis since it is their perception that we are assessing.

We ran a quick search in MEDLINE to learn how previous studies conducted surveys of dental schools and academics. We combined "Faculty, Dental"[Mesh] then (“dental schools” AND surveys) with “global”, “international” and “multi-country” for 2016-2020. The studies we retrieved were mostly about dental students; and included a maximum of four countries. A survey on dental schools focused on dental schools in Europe (https://pubmed.ncbi.nlm.nih.gov/26344938/) and the response rate was 57%. The second survey was a ‘global’ study although there were no dental schools from South America and Africa and not all schools in each country were included (https://pubmed.ncbi.nlm.nih.gov/28423455/). This study had a response rate of 44%. We concluded that multi-country studies about dental schools or dental academics are few, and global studies may have challenges finding a comprehensive framework for sampling. For a global pandemic with a global impact, we felt a convenience sample of dental academics as proxy representatives of dental schools was a suitable alternative. We acknowledged the cons of this method in the study limitations. Despite the limitation of this study, we feel the information provided is needed for the COVID-19 response and relevant for the global dental academic community.

Reviewer 3 Report

It would be interesting to have notion of how many subjects were originated from which source. That is, how many invitations were sent, how many were denied, how many were accessed ... I suggest a flowchart

It is also important to highlight how the adaptation to other languages ​​was so different. Was there just a basic translation? what are the limitations of this?

Details about the data collection form are necessary:
-Describe who made the instrument
-What is the theoretical framework that supported its construction;
-Describe, in addition to your variables and type of questions, how to evaluate the answers
-In the case of the pre-test / validation, how the instrument's validation process took place (content, clarity and objectivity) and which technique was used for consensus in the questions. This is very important since most of these questions have instruments validated in the literature for this subject. In the absence of a validation process, the research is subject to measurement and interpretation views.

Clearly define all results, exposures, predictors, potential confounders and effect modifiers

For each variable of interest, provide data sources and details of the evaluation (measurement) methods.

Describe any efforts to address potential sources of bias

Author Response

We thank the Editor and the Reviewers for the opportunity to revise our paper and the constructive comments to improve it. We provide below a point-by-point response to all comments.

  • It would be interesting to have notion of how many subjects were originated from which source. That is, how many invitations were sent, how many were denied, how many were accessed ... I suggest a flowchart

Response: We thank the Reviewer for this suggestion. We added in the Appendix a new list showing responses by country. (Please, find this list in the attached Word file, as we were not able to insert the table in this text box.)

We distributed the questionnaire through links posted on social media of dental educators’ groups in addition to emails and therefore it was an open survey making it challenging to compute the number of persons reached and the response rate. While we value the importance of documenting response rates, we were limited by the COVID-19 pandemic and schools’ closure which made it difficult to reach peers and we had to use more personal and informal contacts. We did not track the emails of participants to maintain confidentiality. Taking the survey was considered implicit consent to participate so we have only the count of those who submitted the survey. Those who refused to take it were not recorded on the system. 

  • It is also important to highlight how the adaptation to other languages ​​was so different. Was there just a basic translation? what are the limitations of this?

Response: The questionnaire was originally developed in English and translated into Farsi by a co-author from Iran (SM), and into Portuguese by a dentist in the private sector from Brazil who was not further involved in the study. It was then back-translated again to English to compare differences between the English and translated versions to ensure their accuracy. All versions had the same questions with the same options for responses in the same sequence. The translations ensured that the resulting versions used the proper cultural and scientific terms for the local context since they were conducted by dentists/dental academics speaking these specific languages.

Details about the data collection form are necessary:

  • Describe who made the instrument.

Response: The study team developed the survey questions based on guidelines from the WHO, the Centers for Disease Control and Prevention (CDC), and previous research (references 17 to 19). There was no existing tool the study could adapt for use at the time of planning and collecting data for this study. This information has been included in the manuscript.

  • What is the theoretical framework that supported its construction.

Response: We added a part in Methods about the framework used to develop the questionnaire. It was based on the hierarchy of controls developed by the US National Institute for Occupational Safety and Health.

  • Describe, in addition to your variables and type of questions, how to evaluate the answers.

Response: We clarified in Methods how we derived the components of the preparedness score using the principal component analysis (PCA) and how the regression coefficients resulting from the PCA were used to calculate the score with positive values of regression coefficients indicating higher preparedness score and negative values indicating lower preparedness score. We used PCA to identify the constructs in the preparedness items. We used this approach as a better alternative to methods that give a score of 1 to each correct/positive/favorable item implying equal weights to all items in the survey regardless of their contribution to the score. This information is highlighted in yellow in the method section of the manuscript.

  • In the case of the pre-test /validation, how the instrument's validation process took place (content, clarity, and objectivity) and which technique was used for consensus in the questions. This is very important since most of these questions have instruments validated in the literature for this subject. In the absence of a validation process, the research is subject to measurement and interpretation views.

Response: We explained in Methods that the questionnaire was assessed for content validity by five dental academics who were not involved in the study. Their responses were not included in the final data analysis. The content validity index for the tool was 0.99. In addition, we conducted PCA as explained in Methods to identify the preparedness components within the items of the questionnaire thus ensuring construct validity.

  • Clearly define all results, exposures, predictors, potential confounders and effect modifiers

Response: The results of the analysis are explained in the Results section of the paper using the tables and text. We used the term “independent variables” (line 263) instead of “exposures”, or “predictors” based on comments from previous Reviewers to avoid implying causation since we were trying to identify the characteristics of dental schools where academics had a greater perception of institutional preparedness rather than what explains this perceived preparedness following the study aim. Country of residence was considered as a confounder and we adjusted for this in the statistical analysis and explained it in the Methods section of the manuscript (lines 269-271). There were no effect modifiers.

  • For each variable of interest, provide data sources and details of the evaluation (measurement) methods.

Response: The data source for the variables was the responses of participants collected through the questionnaire. We also used an additional source of data for the country level classification of income based on the World Bank data bank. We explained the source and the classification in Methods (lines 265-269).

  • Describe any efforts to address potential sources of bias.

Response: As mentioned in lines 209-210, we suspected that academics with administrative positions may report a higher level of perceived preparedness measures than those not involved in administrative work. Consequently, we added a question in the questionnaire that identified academics with administrative positions and included this in the multilevel analysis. We also addressed the effect of potential confounders by conducting multivariable analysis which was multilevel to account for the clustering of participants within countries. We added this information about potential confounders in our discussion on limitations in the Discussion section of the manuscript.

Reviewer 4 Report

The authors present an interesting study analyzing the differences between dental academic institutions to cope with the COVID1-19 pandemic.

Lines 79-80: please clarify what you mean by “…the attributes of institutions with different degrees of perceived preparedness”. The meaning of attributes is unclear.

Line 94: please state what “LICs” mean

Line 140: the ethical approval date refers to 2016. Since the study and the pandemic occurred in 2020, what study was approved in 2016?

Line 235: table S1 is missing from the uploaded documents

All questionnaires had responses in all items? There is no missing data in some domains?

Lines 329-332: the authors refer that: “Academics who were administrators reported more perceived preparedness measures. This may be biased reporting reflecting their higher concern to ensure patient safety and minimize the possibility of cross-infection [31] in dental clinics during the pandemic in addition to their own perception that they are expected to report more preparedness.” Did the authors consider analyzing separately and compare the responses between the academics who were administrators and the ones that were not?

Author Response

We thank the Editor and the Reviewers for the opportunity to revise our paper and the constructive comments to improve it. We provide below a point-by-point response to all comments.

The authors present an interesting study analyzing the differences between dental academic institutions to cope with the COVID1-19 pandemic.

  • Lines 79-80: please clarify what you mean by “…the attributes of institutions with different degrees of perceived preparedness”. The meaning of attributes is unclear.

Response: We replaced the term “attributes” with “characteristics” so that it would be easier to understand and to match the term we used later in the paper. In the paper itself, in the Methods section (lines 202-210), we explained that section 3 of the questionnaire assessed the institutions’ characteristics and explained what these characteristics were (public or private, teaching, research, or both, undergraduate, postgraduate, or both, number of students and staff, number of patients receiving care ….). These characteristics are assumed to affect the institution’s ability to be prepared and equipped to face the COVID-19 pandemic.

  • Line 94: please state what “LICs” mean

Response: LICs are low-income countries. The expansion of this abbreviation has been added to the abstract. We thank the Reviewer for bringing this point to our attention.

  • Line 140: the ethical approval date refers to 2016. Since the study and the pandemic occurred in 2020, what study was approved in 2016?

Response: This is not the date of IRB approval for the present study. It is the format of ethical approval numbers issued by the Research Ethics Committee in the Faculty of Dentistry, Alexandria University, Egypt. The year 2016 refers to the time the committee began issuing approvals using this format. This study was approved on February 16th, 2020.

  • Line 235: table S1 is missing from the uploaded documents

Response: We thank the Reviewer for raising this issue. We added it after the questionnaire as the second item in the appendix. It includes the number of responses per country as follows:

A2. Countries included in the study (n= 1820 participants)

(Please, find the table in the attached Word file, as we were not able to insert the table in this text box.)

  • All questionnaires had responses in all items? There is no missing data in some domains?

Response: This is correct. The questions were all required to be answered and a participant was free to either respond in full and submit or leave questions unanswered/not complete the survey and not submit it. This was needed since item non-response would have rendered the responses useless because we would not be able to calculate the preparedness score.

  • Lines 329-332: the authors refer that: “Academics who were administrators reported more perceived preparedness measures. This may be biased reporting reflecting their higher concern to ensure patient safety and minimize the possibility of cross-infection [31] in dental clinics during the pandemic in addition to their own perception that they are expected to report more preparedness.” Did the authors consider analyzing separately and compare the responses between the academics who were administrators and the ones that were not?

Response: We included this “academics working as administrators” as a factor in the multilevel analysis (table 3) and therefore adjusted for its effect. However, we did not display separate estimates of the score for those who were administrators and those who were not since it was not one of the aims of the study.

Round 2

Reviewer 3 Report

Thank you very much for accepting the previous suggestions. I think that table 03 should be reorganized. It is necessary in multilevel linear regression analysis to highlight the gross odds and the adjusted odds.

In addition, it would be more interesting to highlight the reference variable, as is done classically, and not to put a variable versus another variable.

Author Response

We thank the Reviewer for the meticulous revision of our work. In response to the Reviewer’s comments, we provide a point-by-point response as follows;

Thank you very much for accepting the previous suggestions. I think that table 03 should be reorganized. It is necessary in multilevel linear regression analysis to highlight the gross odds and the adjusted odds.

  • Response: We reorganized table 3 following the Reviewer’s recommendation. We added the unadjusted and the adjusted estimates. We also added a part in the text to highlight the significant associations (lines 291-294).

In addition, it would be more interesting to highlight the reference variable, as is done classically, and not to put a variable versus another variable.

  • Response: In table 3, we identified the reference category for each variable in a separate row. In table 4, the reference category (HICs) is the same for all rows. To avoid adding a new column headed “HICs” where the term “reference category” is repeated in each row, we clarified in the footnote under the table that HICs is the reference category (line 321). We used this approach to avoid adding a new column to an already busy table.

This manuscript is a resubmission of an earlier submission. The following is a list of the peer review reports and author responses from that submission.

Round 1

Reviewer 1 Report

This manuscript reports the results of a survey of a convenience sample of people working in dental schools in different parts of the world. The reported aim of the survey was to assess the preparedness of dental schools to face the COVID-19 pandemic and to investigate factors that may be associated with preparedness. Taken at face value, this aim seems to be a useful one to address, there are several issues with the approach used and hence the contribution to the scientific literature on the subject:

  • For the results of this work to be useful for the leaders of dentals schools across the world, the rationale and framework for the study need to be better explained. For instance, if schools want to improve preparedness, they need to focus on understanding mutable factors in a study like this. The authors also need to focus on a readily understandable outcome indicator rather than one that is the product of a principal components analysis.
  • For instance, at the beginning of the Discussion section, summarizing the study results, the authors state "The results showed that during the COVID-19, private dental academic 296 institutions and those not having adequate IPC were less prepared for the pandemic and that 297 institutions that were least prepared for the pandemic were more likely to be temporarily closed." People can't change the private/public nature of their institution; having or not IPC is clearly directly related to preparedness, indeed it is part of preparedness (see comments below); and one would expect those institutions that are not prepared would be closed or indeed because they are closed they don't need to be prepared.
  • In short, the framework for the study is not helpful and nor are the results. It would perhaps have been more helpful to perform analyses within particular groups of schools where the environment (i.e. immutable characteristics) are different so like institutions can learn from each other.
  • The authors do not define preparedness. They describe consulting WHO, CDC and other guidelines to create the questionnaire and the approach appears to have involved the creation of a checklist of 15 items that together comprise preparedness. While these items have face validity and were indeed verified for that by 5 dental academics, it is not clear how they fit with a definition of preparedness and how the authors verified the validity of the checklist psychometrically?
  • The 2nd part of the questionnaire the authors developed asks whether institutions have certain items linked with infection prevention and control (IPC). Again, it is not clear how this list was developed and what its validity is. Furthermore it is not clear why this list is not part of the list of preparedness items? Surely having infection control items is part of preparedness?
  • Linked to this, the authors present the second section of the questionnaire, concerning IPC items, as an independent factor in the analyses, with a score ranging from 0-14, depending upon how many of then items schools had. It seems highly likely that this "IPC score" would be linked with preparedness in institutions, so it is not clear that this really is an "independent" factor. In other words it seems these two factors would be correlated so analyzing them in the manner described in the manuscript does not seem very helpful - we expect there to be an association.
  • Furthermore, why did the authors take a very different approach to generating a score for preparedness compared to that of the IPC score. The authors present a complex approach to generate a preparedness score and it is not clear why they did not simply use a 0-15 scale much like they did for the IPC score?
  • In short, the definition and evaluation of preparedness need a clearer explanation, as does the analytic approach.
  • The study design was understandably cross sectional and the authors acknowledge this but they do not explore the implications for the results for readers. As previously alluded to in my comments we do not know the direction of association in their analyses and this can be important if the goal is to inform decision-makers. Also data were collected during the period March to August, during which there were very significant changes both within and between institutions and countries and regions so analyzing data provided by a participant in a closed school in a LIC in March with data provided by a participant from an open institution in a HIC in August has questionable validity
  • Regarding the sample, there are a number of issues:
    • the authors state that they used a convenience and snowballing sampling strategy, which is understandable given the desire to have participants from across the world. However, it is not clear who was targeted as participants - schools/institutions or professors or [non-academic] staff or even students? Given the description of the approach using social media and informal networks, it is highly plausible that a mixture of representatives of schools participated. Even if the participants were only academics, what roles did they have in their schools and to what extent did they know the answers to the questionnaire? The validity of the responses is therefore highly questionable. Did the authors consider this in the analyses?
    • given this situation, it also seems highly plausible that more than one person could have responded for one institution. If this is the case, did they provide the same responses (this could have been used to validate the questionnaire) and how did the authors deal with this in the analyses?
    • The authors mention a sample size estimate but it is not clear why they used the approach described. Firstly, while the aims were clear, there were two different aims, which require two different approaches to sample estimation - one concerning the precision of estimates and the other concerning the strength of association. And how does this fit with the stated null hypothesis of no association?
  • The authors' description of the study's limitations is very........ limited.

Author Response

Response to Reviewer 1’s comments

Point 1: This manuscript reports the results of a survey of a convenience sample of people working in dental schools in different parts of the world. The reported aim of the survey was to assess the preparedness of dental schools to face the COVID-19 pandemic and to investigate factors that may be associated with preparedness. Taken at face value, this aim seems to be a useful one to address, there are several issues with the approach used and hence the contribution to the scientific literature on the subject:

For the results of this work to be useful for the leaders of dentals schools across the world, the rationale and framework for the study need to be better explained. For instance, if schools want to improve preparedness, they need to focus on understanding mutable factors in a study like this. The authors also need to focus on a readily understandable outcome indicator rather than one that is the product of a principal components analysis.

Response 1: We thank the Reviewer for this useful comment. We modified the aim to clarify that we assessed the various aspects of preparedness of dental academic institutions and determined the attributes associated with different degrees of preparedness. In this approach, we focus more on identifying the factors that characterize institutions with -for example- low preparedness so that more support can be directed towards them rather than identifying the factors that can be changed to improve the situation. In addition, we added a definition of preparedness in the Introduction and a framework that explains the factors that we listed under preparedness at the beginning of the Methods section. These factors that constitute preparedness -according to our definition and under the framework- are the ones that need to be addressed to improve preparedness. Because of the extent of the concept of preparedness, its several aspects and the multiple items listed under it, we used PCA to reduce these items. Using a single item would address one aspect only of the multidimensional concept of preparedness.

Point 2: For instance, at the beginning of the Discussion section, summarizing the study results, the authors state "The results showed that during the COVID-19, private dental academic institutions and those not having adequate IPC were less prepared for the pandemic and that institutions that were least prepared for the pandemic were more likely to be temporarily closed." People can't change the private/public nature of their institution; having or not IPC is clearly directly related to preparedness, indeed it is part of preparedness (see comments below); and one would expect those institutions that are not prepared would be closed or indeed because they are closed they don't need to be prepared.

Response 2: Again, we thank the Reviewer for this useful insight. Based on this and other comments, we changed the analysis plan. We removed the availability of IPC items and closure status from the independent variables, in addition to other changes.

Point 3: In short, the framework for the study is not helpful and nor are the results. It would perhaps have been more helpful to perform analyses within particular groups of schools where the environment (i.e. immutable characteristics) are different so like institutions can learn from each other.

Response 3: We strived to do that in the new modified analysis plan by introducing factors that describe the profile of countries (country level income) and academic institutions (whether they were public or private, small or large sized, teaching, research or both), providing under or postgraduate programs and serving a small or large number of patients. We calculated regression estimates for the preparedness reported in each of these types of institutions so that institutions sharing the same attributes would be aware of what others did. 

Point 4: The authors do not define preparedness. They describe consulting WHO, CDC and other guidelines to create the questionnaire and the approach appears to have involved the creation of a checklist of 15 items that together comprise preparedness. While these items have face validity and were indeed verified for that by 5 dental academics, it is not clear how they fit with a definition of preparedness and how the authors verified the validity of the checklist psychometrically?

Response 4: Following these comments, we provided a definition of preparedness in the Introduction and a framework of what constitutes preparedness at the beginning of the Methods section under ‘Questionnaire’. The content validity was assessed, and we reported the content validity index in the Methods. All items which we conceptualized as constituting preparedness were included in the PCA that split them into the 5 areas shown in the Results.

Point 5: The 2nd part of the questionnaire the authors developed asks whether institutions have certain items linked with infection prevention and control (IPC). Again, it is not clear how this list was developed and what its validity is. Furthermore it is not clear why this list is not part of the list of preparedness items? Surely having infection control items is part of preparedness?

Response 5: This was developed based on the reference studies listed below. Its content validity was assessed with the questionnaire with content validity index=0.99. Following the Reviewer’s comment, we included the IPC items with the other preparedness items in the PCA and these items loaded on two components.  

References

Meng, L.; Hua, F.; Bian, Z. Coronavirus Disease 2019 (COVID-19): Emerging and Future Challenges for Dental and Oral Medicine. J. Dent. Res. 2020, 99, 481–487, doi:10.1177/0022034520914246.

World Health Organization Rational use of personal protective equipment for coronavirus disease 2019 (COVID-19)  - Interim guidance; 2020; https://www.who.int/emergencies/diseases/novel-coronavirus-2019/technical-guidance-publications?publicationtypes=01bc799c-b461-4a52-8c7d-294c84cd7b2d

Centers for Disease Control and Prevention Interim Infection Prevention and Control Recommendations for Healthcare Personnel During the Coronavirus Disease 2019 (COVID-19) Pandemic; 2020; Vol. 2; Guidance for Dental Settings | CDC Available online: https://www.cdc.gov/coronavirus/2019-ncov/hcp/dental-settings.html (accessed on Nov 9, 2020).

Point 6: Linked to this, the authors present the second section of the questionnaire, concerning IPC items, as an independent factor in the analyses, with a score ranging from 0-14, depending upon how many of then items schools had. It seems highly likely that this "IPC score" would be linked with preparedness in institutions, so it is not clear that this really is an "independent" factor. In other words it seems these two factors would be correlated so analyzing them in the manner described in the manuscript does not seem very helpful - we expect there to be an association.

Response 6: We removed the IPC score from the list of independent factors and added all its items to the PCA to assess the components of preparedness as reported by the participants. These IPC items loaded on 2 new components: infection control equipment and personal apparel. In addition to the three components previously identified by the PCA.

Point 7: Furthermore, why did the authors take a very different approach to generating a score for preparedness compared to that of the IPC score. The authors present a complex approach to generate a preparedness score and it is not clear why they did not simply use a 0-15 scale much like they did for the IPC score?

Response 7: Now there are no different approaches to create scores since the IPC items are included in the PCA and all factors loaded on 5 components. We didn’t use a preparedness score on a scale from 0 to 29. We used PCA and it generated regression coefficients that were saved into the dataset and used in the statistical analysis so that each item had its weight based on its contribution to the preparedness score instead of all items having equal weights. Using PCA helped in identifying the components of preparedness that were represented by these items.

Point 8: In short, the definition and evaluation of preparedness need a clearer explanation, as does the analytic approach.

Response 8: Based on the Reviewer’s comment, we provided a definition for preparedness in the Introduction, an explanation of the framework that guided the selection of items used in PCA in the Methods/Questionnaire, and also modified the analysis plan to remove factors related to the dependent variable from being used to explain it.

Point 9: The study design was understandably cross sectional and the authors acknowledge this but they do not explore the implications for the results for readers. As previously alluded to in my comments we do not know the direction of association in their analyses and this can be important if the goal is to inform decision-makers. Also data were collected during the period March to August, during which there were very significant changes both within and between institutions and countries and regions so analyzing data provided by a participant in a closed school in a LIC in March with data provided by a participant from an open institution in a HIC in August has questionable validity.

Response 9: We added to the limitation a part about the implication of the study design for the interpretation of findings. We also included a part about the various stages of the pandemic and responsiveness measures in different countries, and how we attempted to account for random variations among countries by including country of residence as random effect variable.

Point 10: Regarding the sample, there are a number of issues: the authors state that they used a convenience and snowballing sampling strategy, which is understandable given the desire to have participants from across the world. However, it is not clear who was targeted as participants - schools/institutions or professors or [non-academic] staff or even students? Given the description of the approach using social media and informal networks, it is highly plausible that a mixture of representatives of schools participated. Even if the participants were only academics, what roles did they have in their schools and to what extent did they know the answers to the questionnaire? The validity of the responses is therefore highly questionable. Did the authors consider this in the analyses?

Response 10: The participant and unit of sampling and analysis was the academic. We added “We used dental academics’ reports to assess institutional preparedness and the unit of sampling and analysis was the academic” to Methods. We also added a clarification that non-academic staff and students were excluded. There was no restriction by specialty of academics or their degrees. We also included administrators and non-administrators as explained. This enabled us to get a more comprehensive perspective. As clarified in the instructions to the survey, we asked participants to report based on their own awareness/ observation. Our study showed a difference between administrators and non-administrators and we explained this by administrators having possibly greater involvement or alternatively being biased to report favorably on their institutions. We added that it is useful to include non-administrators to have a more realistic reporting. Because we asked participants to report on what they observed and whether or not they were administrators then included this in the analysis, we believe that the validity of the responses would not be jeopardized.  

Point 11: given this situation, it also seems highly plausible that more than one person could have responded for one institution. If this is the case, did they provide the same responses (this could have been used to validate the questionnaire) and how did the authors deal with this in the analyses?

Response 11: We could not collect information about the name of the institution to maintain the participants’ confidentiality. It is possible that more than one academic may have responded from the same institution although we had no way of confirming this. We controlled for institutional attributes to accommodate for this issue. We did not report on the frequency of a preparedness measure per institution or per country for this reason too.

Point 12: The authors mention a sample size estimate but it is not clear why they used the approach described. Firstly, while the aims were clear, there were two different aims, which require two different approaches to sample estimation - one concerning the precision of estimates and the other concerning the strength of association. And how does this fit with the stated null hypothesis of no association?

Response 12: The null hypothesis was based on the analytical aim of the study, not the descriptive. We calculated sample size to ensure adequate power for the precision of the estimate. Based on the Reviewer’s comment, we added a justification of the sample in the Methods to show that it also ensured power for the analytical part.

Point 13: The authors' description of the study's limitations is very........ limited.

Response 13: We added discussion to the limitations based on the Reviewer’s comment and hope that this added to it.

Point 14: The Reviewer indicated that ‘Moderate English changes are required’.

Response 14: The entire manuscript has been edited and proof-read for English language accuracy.

Point 15: The Reviewer indicated that ‘improvements must be made to the research design, description of methods, and the presentation of the results.’

Response 15: Substantial improvements have been made to aforementioned parts, as evident by the previous responses and tracked changes in the revised manuscript.

Reviewer 2 Report

Collaborative effort as a response is the lesson from this crisis. This concept is unfortunately absent from the manuscript, albeit being very present in the day-to-day work of the dental academia.

The manuscript reports a very interesting analysis of the preparedness and initial (March-August) response of numerous academic dental institutions to the COVID outbreak.

Despite of the fact that the authors cite the ADEE webpage as a source of information (6), the litterature review has missed some publications on the COVID published (Free access) between March and September 2020 by the European Journal of Dental Education and dedicated to the response of academic dental institutions to the pandemics:

Quinn, B, Field, J, Gorter, R, et al. COVID‐19: The immediate response of european academic dental institutions and future implications for dental education. Eur J Dent Educ. 2020; 24: 811– 814. https://doi.org/10.1111/eje.12542

Liu, X, Zhou, J, Chen, L, Yang, Y, Tan, J. Impact of COVID‐19 epidemic on live online dental continuing education. Eur J Dent Educ. 2020; 24: 786– 789. https://doi.org/10.1111/eje.12569

Bennardo, F, Buffone, C, Fortunato, L, Giudice, A. COVID‐19 is a challenge for dental education—A commentary. Eur J Dent Educ. 2020; 24: 822– 824. https://doi.org/10.1111/eje.12555

Wu, DT, Wu, KY, Nguyen, TT, Tran, SD. The impact of COVID‐19 on dental education in North America—Where do we go next?. Eur J Dent Educ. 2020; 24: 825– 827. https://doi.org/10.1111/eje.12561

Mahendran, K, Yogarajah, S, Herbert, C, Nayee, S, Ormond, M. COVID‐19 and Postgraduate Dental Training—A commentary. Eur. J. Dent. Educ. 2020; 00: 1– 5. https://doi.org/10.1111/eje.12600

Sukumar, S, Dracopoulos, SA, Martin, FE. Dental education in the time of SARS‐CoV‐2. Eur J Dent Educ. 2020; 00: 1– 7. https://doi.org/10.1111/eje.12608

The authors, moreover, haved missed potentially interesting information by not reviewing some Dental Education Associations webpages, such as the ABENO (Brazilian Association for Dental Education) that published in July a very comprehensive guide for a seccure re-opening of the dental hospitals (http://www.abeno.org.br/arquivos/downloads/retomada_de_praticas_seguras_no_ensino_odontologico.pdf) or the ADEA (American Dental Education Association), which offered an open online Member Community COVID-19 response globally open to all Dental Academics; and , again, the abovementionned webpage of the ADEE, whose Executive Committee curated and maintained a dedicated webpage to all the COVID-19 related surveys from sister associations (https://adee.org/members/covid-19-resources/covid-19-related-survey-requests) as well as a webpage including most of the COVID-19 resources made available for the Academic Dental Community (https://adee.org/members/covid-19-resources) related to Global aspects of the COVID pandemics, the Clinical Setting, the Academic Setting and a Discussion Forum.

All the information from these documents and webpages, published in open access and immediately available to the global dental academia, would have improved greatly the analysis on the factors associated with the preparedness of the dental academic institutions to maintain and adapt their clinical and teaching activities to the present and the previsible future pandemic outbreaks.

Author Response

Response to Reviewer 2’s comments

Point 1: Collaborative effort as a response is the lesson from this crisis. This concept is unfortunately absent from the manuscript, albeit being very present in the day-to-day work of the dental academia.

Response 1: We added a part in discussion to clarify the importance of this concept.

Point 2: The manuscript reports a very interesting analysis of the preparedness and initial (March-August) response of numerous academic dental institutions to the COVID outbreak.

Response 2: We thank the Reviewer for this comment.

Point 3: Despite of the fact that the authors cite the ADEE webpage as a source of information (6), the literature review has missed some publications on the COVID published (Free access) between March and September 2020 by the European Journal of Dental Education and dedicated to the response of academic dental institutions to the pandemics:

Quinn, B, Field, J, Gorter, R, et al. COVID‐19: The immediate response of european academic dental institutions and future implications for dental education. Eur J Dent Educ. 2020; 24: 811– 814. https://doi.org/10.1111/eje.12542

Liu, X, Zhou, J, Chen, L, Yang, Y, Tan, J. Impact of COVID‐19 epidemic on live online dental continuing education. Eur J Dent Educ. 2020; 24: 786– 789. https://doi.org/10.1111/eje.12569

Bennardo, F, Buffone, C, Fortunato, L, Giudice, A. COVID‐19 is a challenge for dental education—A commentary. Eur J Dent Educ. 2020; 24: 822– 824. https://doi.org/10.1111/eje.12555

Wu, DT, Wu, KY, Nguyen, TT, Tran, SD. The impact of COVID‐19 on dental education in North America—Where do we go next?. Eur J Dent Educ. 2020; 24: 825– 827. https://doi.org/10.1111/eje.12561

Mahendran, K, Yogarajah, S, Herbert, C, Nayee, S, Ormond, M. COVID‐19 and Postgraduate Dental Training—A commentary. Eur. J. Dent. Educ. 2020; 00: 1– 5. https://doi.org/10.1111/eje.12600

Sukumar, S, Dracopoulos, SA, Martin, FE. Dental education in the time of SARS‐CoV‐2. Eur J Dent Educ. 2020; 00: 1– 7. https://doi.org/10.1111/eje.12608

Response 3: We thank the Reviewer for drawing our attention to these useful resources. We incorporated them in the paper.

Point 4: The authors, moreover, haven missed potentially interesting information by not reviewing some Dental Education Associations webpages, such as the ABENO (Brazilian Association for Dental Education) that published in July a very comprehensive guide for a secure re-opening of the dental hospitals (http://www.abeno.org.br/arquivos/downloads/retomada_de_praticas_seguras_no_ensino_odontologico.pdf) or the ADEA (American Dental Education Association), which offered an open online Member Community COVID-19 response globally open to all Dental Academics; and , again, the above mentionned webpage of the ADEE, whose Executive Committee curated and maintained a dedicated webpage to all the COVID-19 related surveys from sister associations (https://adee.org/members/covid-19-resources/covid-19-related-survey-requests) as well as a webpage including most of the COVID-19 resources made available for the Academic Dental Community (https://adee.org/members/covid-19-resources) related to Global aspects of the COVID pandemics, the Clinical Setting, the Academic Setting and a Discussion Forum.

All the information from these documents and webpages, published in open access and immediately available to the global dental academia, would have improved greatly the analysis on the factors associated with the preparedness of the dental academic institutions to maintain and adapt their clinical and teaching activities to the present and the previsible future pandemic outbreaks.

Response 4: We agree with the Reviewer that these documents and webpages provide valuable information that help guide the dental academic community on how to best deal with the pandemic and how to carry on the academic mission. At the time the study was planned, however, most of these were not available. We started data collection in March 2020 and the survey links have been distributed since that time in various locations. For example, the Brazilian document was released in July, after we started. Some of the ADEA announcements are in May and others are updates in October. We acknowledge the valuable information in these resources and cited them in the paper.

Point 5: The Reviewer indicated that improvements must be made to the ‘introduction to provide sufficient background and include all relevant references.’

Response 5: The introduction has been edited to provide a more extensive background and more relevant references.

Reviewer 3 Report

The study assessed preparedness of academic institutions specialised in dentistry in the combat against COVID-19. The study was based on questionnaires answered by academics from several countries. The article is well written, methods are adequate and results are very interesting.

Minor corrections are needed. The pdf version of the revised manuscript is attached, which includes comments, suggestions and minor corrections using the "comment" tool in Adobe.

Author Response

Response to Reviewer 3’s comments

Point 1: The study assessed preparedness of academic institutions specialized in dentistry in the combat against COVID-19. The study was based on questionnaires answered by academics from several countries. The article is well written, methods are adequate and results are very interesting.

Minor corrections are needed. The pdf version of the revised manuscript is attached, which includes comments, suggestions and minor corrections using the "comment" tool in Adobe.

Response 1: We thank the Reviewer for the valuable comments and suggestions. We modified as indicated using track changes and highlights to show where modifications were made.

Point 2: In response to the Reviewer questions/ suggestions in the pdf:

Which author/institution in Brazil?

Response 2: We specified the name of the author from Iran. We had no collaborator from Brazil. A dentist in the private sector from Brazil helped in the translation. He did not collect data and did not further collaborate as a co-author in the study.

Point 3: Review formatting (Table 1)

Response 3: We aligned the text and modified the spacing so that it would be easier to read in the tables.

Point 4: Or because they are more aware of the institutional infrastructure etc

Response 4: We added this to the paragraph in Discussion.

Point 5: The Reviewer indicated that ‘improvements must be made to the description of methods and the presentation of the results.’

Response 5: Substantial improvements have been made to aforementioned parts, as evident by the tracked changes in the revised manuscript.

Round 2

Reviewer 1 Report

The authors have responded well to previous comments and have improved the clarity of the manuscript. Nevertheless, there remains a fundamental problem which is major - the authors are investigating the preparedness of institutions but they use individual professors as unit of analysis. The sampling strategy focused on individual professors rather than institutions but did not gather information concerning the identity of institutions so they have no idea how many institutions are represented among the sample. Given the recruitment strategy, it seems very likely that there are more than one responses from at least some institutions. If these responses are the same or very similar and there are several of them from several institutions (e.g. 10 institutions with 10 professors responding) then this will skew the principal components analysis. Alternatively, if the answers from different individuals at the same institution are significantly different, then this raises questions concerning the validity of the responses. Indeed, we have no idea how well participants knew the answers to the questions posed, so there are real questions of validity of the responses with no verification. In short, this is a serious flaw with the method, which raises serious questions concerning the validity of the findings.